# Therapeutic Efficiency of Proteins Secreted by Glial Progenitor Cells in a Rat Model of Traumatic Brain Injury

**DOI:** 10.3390/ijms241512341

**Published:** 2023-08-02

**Authors:** Diana I. Salikhova, Victoria V. Golovicheva, Timur Kh. Fatkhudinov, Yulia A. Shevtsova, Anna G. Soboleva, Kirill V. Goryunov, Alexander S. Dyakonov, Victoria O. Mokroysova, Natalia S. Mingaleva, Margarita O. Shedenkova, Oleg V. Makhnach, Sergey I. Kutsev, Vladimir P. Chekhonin, Denis N. Silachev, Dmitry V. Goldshtein

**Affiliations:** 1Institute of Molecular and Cellular Medicine, RUDN University, 117198 Moscow, Russia; tfat@yandex.ru (T.K.F.); annasobo@mail.ru (A.G.S.); margarita.shedenkova@gmail.com (M.O.S.); dvgoldshtein@gmail.com (D.V.G.); 2Research Centre for Medical Genetics, 115478 Moscow, Russia; elemental190@gmail.com (A.S.D.); victoria-mok@yandex.ru (V.O.M.); natalmingaleva@gmail.com (N.S.M.); dv@rm7.ru (O.V.M.); kutsev@mail.ru (S.I.K.); 3A.N. Belozersky Institute of Physico-Chemical Biology, Lomonosov Moscow State University, 119992 Moscow, Russia; viktoriia.golovicheva@yandex.ru; 4Avtsyn Research Institute of Human Morphology of Federal State Budgetary Scientific Institution “Petrovsky National Research Centre of Surgery”, 117418 Moscow, Russia; 5V.I. Kulakov National Medical Research Center of Obstetrics, Gynecology and Perinatology, 117997 Moscow, Russia; yu_shevtsova@oparina4.ru (Y.A.S.); k_gorunov@oparina4.ru (K.V.G.); 6Faculty of Bioengineering and Bioinformatics, Lomonosov Moscow State University, 119234 Moscow, Russia; 7Serbsky State Scientific Center for Social and Forensic Psychiatry, 119034 Moscow, Russia; chekhoninnew@yandex.ru

**Keywords:** secretome, glial cells, iPSCs, proteomic analysis, traumatic brain injury

## Abstract

Traumatic brain injuries account for 30–50% of all physical traumas and are the most common pathological diseases of the brain. Mechanical damage of brain tissue leads to the disruption of the blood–brain barrier and the massive death of neuronal, glial, and endothelial cells. These events trigger a neuroinflammatory response and neurodegenerative processes locally and in distant parts of the brain and promote cognitive impairment. Effective instruments to restore neural tissue in traumatic brain injury are lacking. Glial cells are the main auxiliary cells of the nervous system, supporting homeostasis and ensuring the protection of neurons through contact and paracrine mechanisms. The glial cells’ secretome may be considered as a means to support the regeneration of nervous tissue. Consequently, this study focused on the therapeutic efficiency of composite proteins with a molecular weight of 5–100 kDa secreted by glial progenitor cells in a rat model of traumatic brain injury. The characterization of proteins below 100 kDa secreted by glial progenitor cells was evaluated by proteomic analysis. Therapeutic effects were assessed by neurological outcomes, measurement of the damage volume by MRI, and an evaluation of the neurodegenerative, apoptotic, and inflammation markers in different areas of the brain. Intranasal infusions of the composite protein product facilitated the functional recovery of the experimental animals by decreasing the inflammation and apoptotic processes, preventing neurodegenerative processes by reducing the amounts of phosphorylated Tau isoforms Ser396 and Thr205. Consistently, our findings support the further consideration of glial secretomes for clinical use in TBI, notably in such aspects as dose-dependent effects and standardization.

## 1. Introduction

Traumatic brain injury (TBI) is an acute damaging event with long-term neurological sequelae. The immediate mechanical disruption of the complex brain architecture affects neuronal, glial, and vascular cells, which respond with the massive production of molecules that trigger inflammation, apoptosis, and oxidative stress in surrounding tissues. Chronic neuroinflammation following traumatic brain injury is initiated by the diffuse activation of microglia, reactive astrocytes, and the recruitment of peripheral immune cells, which are exacerbated by the disruption of the blood–brain barrier. The chronic phase may persist indefinitely and spread to intact brain regions, leading to behavioral and cognitive impairments [1,2,3]. The local inflammatory response is accompanied by the activation of matrix metalloproteinases (MMPs). The intense production of MMPs under conditions of acute brain injury impairs the integrity of tight junction proteins (claudin-5, occludin) in the endothelium and undermines the blood–brain barrier [3]. It should be noted that axons and dendrites are particularly affected in all degrees of severity of traumatic brain injury. Alterations in the activity of endogenous signaling pathways in the region of traumatic brain injury lead to the release of Tau protein and its subsequent hyperphosphorylation (p-Tau) [4]. The accumulation of p-Tau leads to the loss of a proper axonal structure and ultimately to the disruption of neuronal integrity. The level of amyloid precursor protein (APP) increases sharply in TBI. APP is located in the somatodendritic zone in axons and in the presynaptic active zone in neurons. An increase in its expression upon damage is a protective mechanism necessary for neuron survival. It is also involved in the mechanisms of neuron development, migration, and synaptogenesis. APP is required for the efficient response of immune and glial cells to inflammatory stimuli. It interacts with various signaling cascades to prevent neurodegeneration and regulates the endosomal phosphoinositide signaling pathway [5].

The overall complexity of the sequelae of TBI, with the simultaneous involvement of multiple interrelated pathogenic processes, requires advanced therapies that target multiple pathogenetic factors simultaneously. A possible approach would be to use stem/progenitor cells. However, stem cell transplantation carries high risks due to adverse side effects. The use of paracrine factors instead of whole-cell products could provide similar benefits while avoiding the limitations of cell transplantation [6,7]. Numerous data have shown that the molecular weight of most secreted proteins regulating neuronal plasticity and inflammation is less than 100 kDa. Such proteins can cross the blood–brain barrier [8,9,10,11] and reach therapeutically effective concentrations in the brain via intranasal administration [12,13]. Glial cells are known to provide structural and metabolic support to brain neurons. They can influence synaptic processes and neuronal network formation and are involved in the regulation of memory and complex behavioral processes [14,15]. Therefore, this work was devoted to studying the neuroprotective effects of the intranasal administration of a protein complex (PC) with a molecular weight (Mw) of 5–100 kDa secreted by glial progenitor cells (GPCs) in a rat model of TBI.

## 2. Results

### 2.1. The Composition of GPC-Secreted Proteins

The resulting fraction contained proteins below 100 kDa, with a pronounced band in the 25 kDa region (Figure 1A), which was confirmed by SDS-PAGE. The fraction of secreted proteins with Mw 5–100 kDa included 142 proteins (Appendix A), which were further categorized as molecular functions (MF), cellular components (CC), and biological processes (BP) using https://string-db.org/ (accessed on 20 July 2023).

Most of the identified proteins were located in the extracellular region (62% of the detected proteins), secretory granules (26%), and extracellular matrix (12%) according to the CC category, suggesting a PC enriched with secreted proteins. However, proteins of intracellular components belonging to the groups “cytosol” (50%), “endomembrane system” (39%), and “cytoplasmic vesicle” (31%) were also identified in this category. There were also membrane proteins contained in the groups “cell junction” (23%) and “cell surface” (10%) (Figure 1B). The molecular function (MF category) of the secreted proteins consisted of “protein binding” (52%), “catalytic activity” (41%), “enzyme and RNA binding” (21%), “protein-containing complex binding and enzyme regulator activity” (14%), and “protein homodimerization and peptidase activity” (11%) (Figure 1C).

According to the BP category, 64% of the proteins were involved in “metabolic processes”, approximately 30% in “immune system processes and immune response”, around 13–15% in “negative regulation of cell death/apoptosis”, and 12% in “oxidation-reduction process” (Figure 1D).

Several components identified by proteomic analysis may have specific importance in TBI; one of them is a member of the 14-3-3 family proteins (YWHAZ). The 14-3-3 proteins have multifaceted functionalities, implicated in apoptosis, cell cycle regulation, and cytoskeleton remodeling. By interacting with intermediate filaments, these proteins can modulate the assembly and disassembly of cytoskeletal elements [16].

Moreover, the PC contained the heat shock proteins; one of them is the heat shock cognate 71 kDa protein (HSPA8, also known as heat shock 70 kDa protein 8). HSPA8 can be actively released from intact cells in a free form and prevents apoptotic processes [17,18,19]. Other anti-apoptotic proteins were presented, including apolipoprotein A-I (APOA1) [20], apolipoprotein D (APOD) [21,22], gremlin (GREM1) [23], filamin-A (FLNA) [24,25], and serpin B3 (SERPINB3) [26].

The proteomic analysis also revealed specific proteins with anti-inflammatory and immunomodulatory properties, including antithrombin III [27,28], galectin-1 (LGALS1), cathepsin D (CTSD), and macrophage migration inhibitory factor (MIF), implicated in immune response regulation and tissue remodeling [29,30,31,32,33].

The mechanisms of cell death in TBI include the formation of reactive oxygen species causing oxidative stress [34,35]. The PC included several proteins that enhanced neuronal survival under oxidative stress conditions, notably peroxiredoxin-1 (PRDX1) [36,37], peroxiredoxin-4 (PRDX4) [38,39], and dermcidin (DCD) [40] (Appendix A).

The variety of biologically active proteins identified in the PC suggests pleiotropic potential, aimed at reducing the inflammatory and apoptotic responses. It is possible that the PC will influence the metabolic processes of neurons and support their structures and functions.

### 2.2. Assessment of Damage Volume

The modeling of TBI led to pronounced damage to the sensorimotor cortex of the brain according to T2-weighted images. The damage volume constituted 70.41 ± 16.53 mm^3^ in the control group and 73.13 ± 4.99 mm^3^ in the experimental group, without statistically significant differences in this parameter on the 14th day after TBI induction (Figure 2).

### 2.3. Neurological Outcomes

The animals revealed no neurological deficits, scoring 14 ± 0.5 points in the limb-placing test prior to TBI modeling. Sham-operated rats during the experiment showed the same score as animals before TBI. Damage to the sensorimotor cortex led to pronounced neurological disturbances on the first day after the injury; the animals in both groups scored 2.4 ± 0.4 points. The treatment afforded significant improvements detected on days 7 and 14 post-TBI as compared with the control group (*p* < 0.05, Figure 3A).

The cylinder test measured the asymmetry of the use of the forelimbs to explore the enclosed space’s walls on day 14 post-TBI. Normally, rats use both forelimbs equally [41]. Sham-operated rats used their contra- and ipsilateral limbs equally to study the walls of the closed space of a glass cylinder. The asymmetry resulted from the unilateral damage to the sensorimotor cortex: the animals made less use of the limb controlled by the damaged hemisphere and were less likely to use both limbs simultaneously. The TBI modeling led to the significantly decreased use of the contralateral limbs to 27%, and the use of the ipsilateral limbs increased to 73%. PC treatment significantly recovered the asymmetry of the forelimb use in rats: up to 60% and 40% for the ipsi- and contralateral limbs, respectively (Figure 3B).

A treatment-related tendency toward enhanced motor coordination and balance as measured by the rotarod test was observed on day 14 post-TBI (248.01 ± 47 s vs. 186.5 ± 86 s in the control group, *p* = 0.141; Figure 3C). Sham-operated rats lasted for the entirety of the allotted time, 290 ± 10 s.

### 2.4. Histological Analysis of Brain Tissue

The immunohistochemical analysis was performed on day 14 post-TBI. The accumulation of reactive astrocytes was detected in both groups around the periphery of the damage area. The largest number of them was observed in the control group—952 ± 290 GFAP^+^ cells per mm^2^. The intranasal administration of the PC decreased the number of reactive astrocytes 2.5-fold compared with the control group (*p* < 0.05, Figure 4). At the same time, significant differences were not observed in the number of astrocytes in the hippocampus and striatum (Appendix A).

CD68^+^ phagocytic macrophages and CD206^+^ M2-polarized macrophages were identified in the damaged area of the cortex but absent in the striatum and hippocampus. Moreover, a decrease in the number of CD68+ cells was not observed in the experimental group. However, significantly higher counts of pro-regenerative M2-polarized macrophages were detected in the same area (three-fold, *p* < 0.05 compared with the control) (Figure 4).

The accumulation of microglial cells was revealed in the striatum and hippocampus, but only single cells were seen at the site of cortex injury (Appendix A). Differences in the number of microglial cells were not detected in these areas of the brain (Appendix A).

### 2.5. Gene Expression Assay

The expression of genes associated with apoptosis, extracellular matrix degradation, and the inflammatory response was analyzed on days 7 and 14 post-TBI. The relative content of pro-apoptotic BAX and anti-apoptotic B-cell lymphoma 2 (BCL2) proteins in a cell determines its sensitivity to apoptotic signals: a cell with a higher BAX/BCL2 ratio will have increased sensitivity to apoptosis [42]. Significant changes in the *Bax*/*Bcl2* mRNA ratio were observed in the PC-treated group in the cortex damage area on day 7 post-TBI. The ratio of *Bax/Bcl2* genes decreased in 2.4-fold in the experimental group (*p* < 0.05; Figure 5A). This therapy did not lead to a significant change in the *Bax/Bcl2* mRNA ratio in the striatum (Figure 5B) and hippocampal area (Appendix A).

The disruption of brain tissue integrity facilitated to increase the expression of matrix metalloproteinases (MMPs) and their tissue inhibitors (TIMPs), which have a variety physiological and pathological functions, including the degradation of the extracellular matrix, the regulation of cytokines/chemokines, and the cleavage of surface receptors [43]. The PC treatment served to decrease *Mmp9* (9-fold) and *Timp1* (4.6-fold) expression in the damaged cortex area, and *Mmp2* (6.4-fold) and *Timp1* (9.6-fold) expression in the striatum, on the seventh day of TBI (*p* < 0.05) (Figure 5A,B). Similar changes were not observed in the hippocampus (Appendix A).

Damaged brain parenchyma facilitates the massive release of signaling molecules that initiate inflammation [44]. The increased expression of some inflammatory marker genes was observed on the seventh day of TBI. The intranasal administration of the PC caused a more than three-fold decrease in the mRNA levels of *Il12a, Il12b, Il18, Il6,* and *Tnfa* in the cortex damage area and *Il18, Il6, Il23, Il1b,* and *Tnfa* in the striatum vs. controls (*p* < 0.05; Figure 5A,B). There was no trend towards a decrease in the expression of pro-inflammatory cytokine genes in the hippocampus (Appendix A). It should be noted that changes in the expression of the studied genes were not observed between the two groups on the 14th day of TBI, except in the cortex damage area (Appendix A). In this area, the expression of the *Il6* gene was significantly reduced 5.4-fold in the experimental group vs. the control group (Appendix A).

### 2.6. Tissue Levels of Active Caspase-9

Damage after TBI leads to the death of neurons. Caspases activated by proteolytic cleavage are the main mediators of apoptotic cell death. Caspase-9 is one of the key effector enzymes in neuronal apoptosis. Based on the data of the PCR analysis, the levels of active caspase-9 were assessed in the striatum and injury site of the cortex. The level of active caspase-9 was three-times lower in the experimental group vs. controls in the cortex damage zone (*p* < 0.05; Figure 6B). No differences were found in the striatum.

### 2.7. Assessment of Neurodegeneration Markers

Neurodegenerative processes can occur after TBI, and their coverage correlates with the severity of the injury [4]. The levels of neurodegeneration associated with APP, Tau, and p-Tau proteins were assessed on day 14 post-TBI. The PC treatment caused a 1.7-fold increase in the level of APP in the cortex damage area vs. the control group, which was probably associated with a protective mechanism necessary to increase the survival of neurons (*p* < 0.05). There were no significant differences in the APP protein levels in the striatum and hippocampus (Figure 7).

The PC treatment also afforded a three-fold enhancement in the Tau protein level in the striatum vs. controls (*p* < 0.05; Figure 7). In other areas of the brain, the level of Tau protein was higher than in the control group but did not reach statistical significance. Hyperphosphorylation of the Tau protein (p-Tau) in TBI undermines axonal transport, leading to axon degradation and eventually causing neuronal death [45]. The ratios of the phosphorylated forms of Tau at Ser396 and Thr205 normalized to the unphosphorylated form were measured to assess the contribution of Tau hyperphosphorylation. The levels of p-Tau (Ser396)/Tau and p-Tau (Thr205)/Tau were more than five-fold decreased after PC treatment in the striatum vs. the control group (*p* < 0.05; Figure 7). For other studied brain regions, we observed a tendency towards decreased levels of p-Tau (Ser396)/Tau and p-Tau (Thr205)/Tau, but the values were not statistically significant (Figure 7).

## 3. Discussion

Glial cells are structural components of the brain that are closely related to neural networks and memory formation and affect the functional activity of neurons. However, the role of glial cells in the functional recovery of the brain after injury has not been elucidated. This cell type can influence different links in pathogenesis after injury by secreting various factors and regulatory molecules not exceeding Mw 100 kDa [46,47]. For example, glial cells can interact intensively with neurons by enhancing neuroplasticity and neuroprotection, or with immune cells by reducing the inflammatory response following traumatic brain injury.

According to the proteomic analysis, the secretome derived from GPCs includes growth factors, neurotrophins, cytokines, enzymes, heat shock proteins, and regulatory factors. The identified proteins are associated with metabolic processes, immune system processes and immune responses, the negative regulation of cell death/apoptosis, and the oxidation-reduction process. Among proteins with anti-apoptotic functions, we can mention 14-3-3 family proteins [16], apolipoprotein A-I (APOA1) [20], apolipoprotein D (APOD) [21,22], gremlin (GREM1) [23], filamin-A (FLNA) [24,25], and serpin B3 (SERPINB3) [26], and proteins implicated in immune response regulation include antithrombin III [27,28], galectin-1 (LGALS1), cathepsin D (CTSD), and macrophage migration inhibitory factor (MIF) [29,30,31,32,33]. All these proteins may be important in the treatment of TBI. Moreover, the intranasal administration of such proteins, as shown by many data, can achieve targeted delivery to the brain [13,48]. This method of administration of the PC promoted sensorimotor recovery but without reducing the damage volume in the rat brain. The obtained result may be associated with the delayed PC treatment after 24 h, when the damage volume was formed. It should be noted that brain injuries lead to characteristic morphological changes in the form of gliomesodermal scars composed predominantly of reactive astrocytes, whose number was reduced in the experimental group at day 14 post-TBI. Such scars effectively isolate the site of injury from intact cortical areas, thereby preventing the spread of the inflammatory reaction. On the other side, neurogenesis and neurorecovery processes are often suppressed in the gliomesodermal scar area, because the scar forms a structural barrier that constantly releases axon growth inhibitors [1,49,50]. It is likely that the action of the PC led to structural changes in brain tissue by reducing the number of reactive astrocytes in the gliomesodermal scar, thereby improving the plasticity of the brain.

TBI disrupts the balance between pro-apoptotic BAX and anti-apoptotic BCL-2, a survival-promoting protein that binds to BAX, which leads to an apoptotic response, as observed in the animals in the control group. PC administration promoted a reduction in the mRNA levels of the *Bax/Bcl* ratio, and, moreover, decreased the level of active caspase-9, which initiated the apoptotic pathway of cell death. Such changes were demonstrated at the site of injury in the acute phase of TBI, but did not affect the striatum and hippocampus. These differences can be explained by the uneven distribution of anti-apoptotic factors across brain regions [12], and treatment during the acute phase.

The mechanical damage of neurons and the loss of brain tissue integrity trigger a neuroinflammatory response, which may spread across the brain with devastating consequences, as well as increasing the expression of matrix metalloproteinases. The PC includes proteins that can be aimed at reducing the inflammatory response by suppressing pro-inflammatory and activating anti-inflammatory factors. Moreover, the PC contains plasminogen activator inhibitor type-1 (PAI-1), which regulates tissue homeostasis and wound healing by inhibiting MMP activation [51]. A significant reduction was observed in markers of extracellular matrix degradation factors (*Mmp9*, *Timp1, Mmp2*) and pro-inflammatory cytokines (*Il18*, *Il23*, *Il12a*, *Il12b*, *Il6*, *Il1b*, *Tnfa*) in the cortex and striatum after PC treatment vs. the control group on the seventh day post-TBI. The lack of differential gene expression by day 14 was consistent with the absence of changes in phagocytic macrophage and microglial cell counts and may reflect the confinement of the positive effect to the acute inflammatory phase of TBI. However, changes in the number of M2-polarized macrophages (CD206^+^-cells) were observed in the chronic phase of TBI. The count of these cells was three-times higher in the cortex damage area under PC treatment. These data indicate the presence of components in the PC with the ability to activate a regenerative phenotype in macrophages.

It is known that the consequences of TBI can lead to neurodegenerative changes in brain tissue. Moreover, neurodegenerative processes can also affect distant regions of the brain during injury. Under normal conditions, the Tau protein is a neuronal growth stimulator and a regulator of axonal transport, and it provides support for microtubule stability. The active phosphorylation of the Tau protein promotes its aggregation, with the formation of oligomers and their exit from microtubules, which leads to their disorganization. p-Tau protein aggregates are involved in the formation of neurofibrillary tangles, which disrupt the functions of synapses. This leads to damage to the axons and the apoptosis of the neurons [45,52]. The administration of the PC decreased the p-Tau level (Ser396 and Thr205) in the chronic phase of TBI, thereby contributing to the maintenance of the neuronal structure in the striatum, but not in other regions of the brain. An increase in the levels of APP is a consequence of the loss of the integrity of the brain tissue, and it acts as a mechanism for the stimulation of synaptogenesis and supports neuron survival in TBI [5]. The PC treatment promoted an increase in APP levels in the cortex in the chronic phase of TBI, thereby improving the functional state of the animals.

It is noteworthy that the changes regarding the intranasal administration of the PC affected only two regions of the brain—the cortex and striatum—and did not influence the hippocampus. It is likely that not all functional abilities of the PC were studied: in addition to proteins with neuroprotective and anti-inflammatory effects, the PC contains proteins that regulate metabolic and oxidation-reduction processes. However, this PC function requires further research. Nonetheless, some therapeutic effects of the PC and the pathogenesis links that it affects were observed in this study. The therapeutic efficiency of the PC was confirmed as it decreased the inflammatory and apoptotic processes, prevented neurodegenerative processes, and improved brain plasticity.

Given the absence of similar works, the obtained results were indirectly compared with works devoted to the study of the therapeutic effects of secretomes obtained by culturing various types of stem/progenitor cells. Many researchers have noted an improvement in the sensorimotor deficit and functional recovery of animals in rodent models of TBI with the administration of secretomes by mesenchymal stem cells (MSCs) obtained from different sources [53,54,55,56,57]. In addition, a therapy based on conditioned media MSCs reduced vasogenic cerebral edema [56] and prevented the development of neurodegenerative processes [55] and apoptosis [53,56]. MSC secretomes also had significant immunomodulatory effects, reducing the expression of pro-inflammatory cytokines (IL-6 and TNF-α) and increasing anti-inflammatory cytokines (TGF-β). At the same time, most researchers have observed the activation of immune cells to the M2 phenotype (an increase in Arg1^+^ cells) and the suppression of the M1 phenotype (a decrease in NOS2^+^ cells) at the site of injury [56].

## 4. Materials and Methods

### 4.1. PC Preparation

The protein complex was derived from GPCs differentiated from human induced pluripotent stem cells (iPSCs) obtained and validated as described previously [58]. The procedure was approved by the Institutional Ethics Committee of the Federal State Budgetary Institution “Research Centre for Medical Genetics” (Protocol No. 2019-2/3 from 13 October 2020). The GPCs (15 passages) were maintained in DMEM/F12 (PanEco, Moscow, Russia) with 15 mM HEPES (PanEco), 1 mM glutamine (PanEco), 1% non-essential amino acid mix (PanEco), 100 mg/L penicillin–streptomycin (PanEco), 1% N2 (PanEco), 1% fetal bovine serum (Gibco, Carlsbad, CA, USA), 20 ng/mL EGF (Peprotech, Cranbury, NJ, USA), and 20 ng/mL CNTF (Peprotech) on a Matrigel (Corning, Glendale, AZ, USA) substrate. The GPCs were cultivated in humidified CO_2_ incubators at 37 C and 5% CO_2_. At 85–95% confluency, the cells were detached with 0.05% Trypsin–EDTA (Capricorn Scientific, Ebsdorfergrund, Germany), pelleted by centrifugation at 1400 rpm for 5 min, and replated in a 2-fold dilution into new T175 flasks (SPL Life Sciences, Pocheon, Republic of Korea). To obtain the conditioned medium, GPCs were washed repeatedly with Hanks’ salt solution (HSS) and cultured in indicator-free DMEM/F12 medium with 15 mM HEPES, 1 mM glutamine, 1% essential amino acid mix, and 100 mg/L penicillin–streptomycin (all by PanEco) for 16 h. The medium was collected, centrifuged from debris at 3000 rpm for 5 min, and filtered sequentially through 0.45 µm and 0.22 µm filters using benchtop vacuum holders (Millipore, Darmstadt, Germany). Proteins of Mw ≥ 100 kDa were eliminated by tangential ultrafiltration in a Labscale TFF System (Millipore) with a Pellicon XL 100 kDa cartridge. The DMEM/F12 medium was replaced with phosphate-buffered saline (PBS) in diafiltration mode. The concentration of the protein solution was performed using a Pellicon XL 5 kDa cartridge. The PC with a molecular weight of 5–100 kDa at a concentration of 250 μg/mL was stored at –80 °C. The quantity of total protein was measured by the Quick Start Bradford Protein Assay (Bio-Rad, Hercules, CA, USA), with bovine serum albumin (Bio-Rad) as a calibration standard. Polyacrylamide gel electrophoresis (SDS-PAGE) was performed using the Bio-Rad system to confirm the size of the proteins. Gels were stained with Coomassie Stain (Bio-Rad) for 2 h at room temperature and rinsed with water, according to the manufacturer’s instructions.

### 4.2. Proteomic Analysis

The conditioned medium cleared from the Mw ≥ 100 kDa fraction was freeze-dried. The obtained lyophilisate was dissolved in 4mL of resuspension buffer containing 10 mM Tris–HCl pH 8.0, 4% SDS, 2 mM DTT for 30 min at 60 °C, and then centrifuged at 16,000× *g* (4 °C) for 10 min. The supernatants were ultrafiltrated using 5 kDa units (Spin-X UF6, Sigma-Aldrich, St. Louis, MO, USA) to reduce the sample volume. Proteins were precipitated with acetone: 900 µL of cold acetone (-20 °C) was added to 100 µL of the sample, and it was stored at −20 °C for 14 h. Then, the samples were centrifuged at 15,000× *g* (4 °C) for 10 min.

The obtained precipitate was dissolved in 1 mL of cold acetone (−20 °C) and centrifuged at 15,000× *g* (4 °C) for 10 min. The dry pellets were dissolved in 20 µL of lysis buffer (100 mM Tris–HCl pH 8.0, 4% SDS, 100 mM DTT) and the samples were heated at 60 °C for 30 min prior to the Quick Start Bradford Protein Assay (Bio-Rad). The analysis involved 30 µg of total protein from each sample. The trypsinolysis was carried out for 14 h at 37 °C; the products were desalinated using SDB-RPS microcolumns, vacuum-dried, and stored at –80 °C until liquid chromatography with tandem mass spectrometry (LC-MS/MS) analysis.

The peptides were separated on Aeris 2.6 µm Peptide XB-C18 columns (Phenomenex, Torrance, CA, USA) using the Ultimate 3000 Nano LC System (Thermo Fisher Scientific, Waltham, MA, USA) coupled to a Q Exactive HF instrument (Thermo Fisher Scientific) with a Nanospray ion source (Thermo Fisher Scientific). For higher accuracy, a list of peaks was generated and subsequently analyzed in the MASCOT 2.5.1 and X! Tandem ALANINE 2017.02.01 pipelines using the UniProtKB database, taxon *H. sapiens*. The resulting files were loaded into Scaffold 4 version 4.0.7 for validation and meta-analysis. All identifications with a local false discovery rate of <0.05 were considered reliable. The functional profiles were analyzed at https://string-db.org/ (accessed on 20 July 2023).

### 4.3. Animals

The experiment involved adult male Wistar rats, with 300–350 g body weight. All manipulations were approved by the Ethical Committee at Avtsyn Research Institute of Human Morphology of FSBSI “Petrovsky National Research Centre of Surgery” (Protocol No. 38(14) from 31 May 2022) and were carried out in accordance with Directive 2010/63/EU of the European Parliament and with the Council and ARRIVE guidelines. The animals were kept under a 12:12 light cycle and 22 ± 2 °C with ad libitum access to food and water.

### 4.4. TBI Modeling

The injury was modeled by dosed contusion damage to the open brain. During all manipulations, rats were anesthetized with mask anesthesia using isoflurane (1.5–2% of air supply); the body temperature was maintained at 37 ± 0.5 °C using a heating pad. The animal was placed in a stereotaxic frame, the parietal surface was shaved, and a median longitudinal incision was created in the skin. A hole was drilled in the skull above the sensorimotor cortex with a 5 mm diameter burr, centered 2.5 mm lateral and 1.5 mm caudal to the bregma. A cylindrical impactor was inserted into the hole to a depth of 3 mm below the dura mater. Injury was caused by a 50 g load falling on the firing pin from a height of 10 cm [59].

### 4.5. Animal Study Design

The animals were randomly divided into 3 groups: a sham-operated group (sham, n = 6); the experimental group with TBI (n = 15), which received 30 µL intranasal infusions of the PC; and the control group with TBI (n = 15), which received 30 µL PBS (Figure 8). Treatment was given daily for 6 days, starting 24 h after traumatic brain injury; the observation period was 2 weeks. The neurological status of the animals (n = 10 per group) was assessed by using the limb-placing test, rotarod, and cylinder tests. MRI scans were performed (n = 10 per group) on day 14 post-TBI. Animals were sacrificed by inhalation anesthesia with a lethal dose of isoflurane for immunohistochemistry, immunoblotting, and molecular genetic analysis on day 7 or 14 post-TBI (n = 5 per group for each method).

### 4.6. MRI Morphometry

MRI was performed in a BioSpec 70/30 scanner (Bruker, Berlin, Germany) with a magnetic field induction of 7 T and a gradient system of 105 mT/m. Animals were anesthetized with isoflurane (5% in an air mixture during administration and 1.5–2% during maintenance) and placed in a positioning device equipped with a stereotaxis and thermoregulation system. The brain damage volume was measured using high-resolution T2-weighted images using the ImageJ 1.8.0 software (Wayne Rasband, National Institute of Mental Health, Bethesda, MD, USA). The damage area was outlined manually. The injury volume was calculated as V = (S1 +...+ Sn) × (h + d), where S1, …, Sn—cut areas, mm^2^; h—slice thickness, mm; d—inter-slice interval, mm.

### 4.7. Behavioral Tests

Specialized equipment from OpenScience LLC (Moscow, Russia) was used for all tests. Sensorimotor recovery was assessed with the limb-placing test, which consisted of 7 tests measuring the functional responses of the forelimbs and hindlimbs to tactile, proprioceptive, and visual stimuli. Neurological status was scored in points: 2, task completed; 1, task completed with a delay of more than 2 s and/or incomplete; 0, task failed [60].

The asymmetry of forelimb use was assessed with the cylinder test. The animal was placed in a 30-cm-tall glass cylinder with a diameter of 20 cm. Movements were recorded for 8–10 min with a video camera pointed at a mirror located at a 45° angle behind the cylinder. The time used in the spontaneous exploration of the cylinder walls was recorded in a frame-by-frame playback. Asymmetry was calculated as (contr + ½ × sim)/(ipsi + sim + contr) × 100, where “contr”—contralateral limb (damaged), “sim”—simultaneous use of the forelimbs, “ipsi”—ipsilateral limb [61].

The rotarod test assessed motor coordination and body balance functions. The animals were placed on a rotating rod with a diameter of 10 cm. The rotating speed was increased linearly from 4 to 40 rpm for 3 min. The test lasted 5 min or until the rat fell off the apparatus. Each animal underwent three training trials on days 11, 12, and 13 post-TBI. The final test was performed on day 14 [62].

### 4.8. Immunohistochemistry

The tissue samples were coated with Tissue-Tek OCT Compound (Sakura, Torrance, CA, USA) and stored at −80 °C. The 8–10 µm cryosections were prepared in a Leica CM1950 cryostat (Leica Biosystems, Nubloch, Germany), transferred to slides coated with Superfrost Plus (Thermo Fisher Scientific), dried, and fixed with cold 4% formaldehyde (Merck KGaA, Darmstadt, Germany) for 10 min. The sections were incubated with primary antibodies against CD68 (1:100, ab125212), CD11b (1:200, ab1211), CD206 (1:500, ab64693), and GFAP (1:600, ab7260) (Abcam, Cambridge, UK) in PBS containing 0.3% Triton X-100 and 2% bovine serum albumin (PanEco) at +4 °C overnight. Then, sections were rinsed and incubated with anti-rabbit IgG (H+L) cross-adsorbed secondary antibody Alexa Fluor™ 488 (1:600, A-11008) or anti-mouse IgG (H+L) cross-adsorbed secondary antibody Alexa Fluor™ 555 (1:600, A-21422) (Thermo Fisher Scientific) for 2 h in the dark. Nuclei were counterstained with DAPI solution (2 µg/mL, Sigma-Aldrich, St. Louis, MO, USA). After this, sections were washed and mounted in Aqua-Poly/Mount (Polysciences, Warrington, PA, USA) under coverslips.

The microscopy was carried out with an Axio Observer.D1 inverted fluorescent microscope equipped with AxioCam HRc (Carl Zeiss, Oberkochen, Germany). The positively stained cells were counted in 8–10 randomly selected fields of view within a 1 mm^2^ brain section area for each sample using ImageJ.

### 4.9. Immunoblotting

The tissue samples were lysed with a Protein Solubilization Buffer Kit (Bio-Rad) and mixed with a protease and phosphatase inhibitor cocktail (Thermo Fisher Scientific). The lysates were centrifuged at 20,000× *g* and 4 °C for 30 min. The supernatants were collected and the pellet was centrifuged again. Then, the supernatants were pooled, and the protein concentration was measured by the Bradford method. Laemmli Sample Buffer 2x (Bio-Rad) containing 5% β-mercaptoethanol (Bio-Rad) was added to the obtained supernatants. Samples were incubated at 99 °C for 10 min before performing Western blotting. The proteins were separated by SDS-PAGE using the TGX Stain-Free FastCast Acrylamide Starter Kit, 10% (Bio-Rad) for neurodegeneration marker analysis and the TGX Stain-Free FastCast Acrylamide Starter Kit, 12% (Bio-Rad) for caspase-9 analysis. Protein transfer was carried out on nitrocellulose membranes using the Trans-Blot Turbo Transfer System (Bio-Rad). The membranes were pre-blocked overnight with 3% BSA in PBS and incubated with primary antibodies to p-Tau Ser396 (1:1000, ab32057), p-Tau Thr205 (1:1000, ab254410), APP (1:1000, ab32136) (all by Abcam, UK), Tau (1:1000, 46687) (Cell Signaling, Danvers, MA, USA), and β-actin (1:1000, 8457) (Cell Signaling) for neurodegeneration marker analysis and β-actin (1:2000, A1978) (Sigma-Aldrich) and cleaved-CASP9 (D353) antibody (1:500, CSB-PA000010) (Cusabio, Wuhan, China) for caspase-9 analysis in PBS with 0.1% Tween-20 and 1% BSA. After this, the membranes were washed and incubated with anti-rabbit IgG (H+L) cross-adsorbed secondary antibody Alexa Fluor™ 488 (1:1000, A-11008) or anti-mouse IgG (H+L) cross-adsorbed secondary antibody Alexa Fluor™ 555 (1:1000, A-21422) (Thermo Fisher Scientific) for 60 min in the dark. The immunoblots were documented using the BioRad Chemidoc Imaging System (Biorad). The semi-quantitation calculation of proteins was carried out in the ImageLab 5.0 software (Bio-Rad) with normalization to β-actin.

### 4.10. Reverse Transcription PCR Assay

The expression of specific genes in brain tissue was measured on days 7 and 14 post-TBI. Several brain areas of the ipsilateral hemisphere were sampled, including the cortex at the injury site, hippocampus, and striatum. The samples were placed in RNAlater (Thermo Fisher Scientific) for RNA stabilization. Total mRNA was isolated using the RNeasy Mini Kit (Qiagen, Venlo, Netherlands) and cDNA synthesis was carried out with the RevertAid™ First Strand cDNA Synthesis Kit (Thermo Fisher Scientific). The real-time polymerase chain reactions were set with qPCRmix-HS SYBR mastermixes (Evrogen, Moscow, Russia) and conducted in Bio-Rad iQ cycler (Bio-Rad). The program used initial denaturation at 95 °C for 5 min followed by 40 cycles of denaturation (95 °C, 20 s), primer annealing (56–63 °C, 20 s), and elongation (72 °C, 20 s). The data were normalized to averaged values for two reference genes, GAPDH and ACTB, using the ΔC (T) method. The gene-specific primers (Appendix A) were designed in NCBI Primer-Blast.

### 4.11. Statistics

Data were analyzed in SigmaPlot 12.5 (Systat Software, San Jose, CA, USA). Normality was checked with the Shapiro–Wilk test. The MRI, RT-PCR, immunoblotting, and immunohistochemistry data were compared using a *t*-test for normal distributions or the Mann–Whitney test for distributions other than normal. Cylinder test data were analyzed by ANOVA on ranks with Dunn’s post hoc test. Limb-placing test data were analyzed by two-way ANOVA with the Holm–Sidak post hoc test for multiple comparisons. Rotarod test data were analyzed by one-way ANOVA with the Holm–Sidak post hoc test for multiple comparisons. Differences with *p* < 0.05 were considered significant.

## 5. Conclusions

Despite the uniqueness of the use the size-fractionated glial cell secretome to promote post-TBI nervous tissue recovery, this study presents a broader concept of the application of conditioned cell culture media in therapy. Thus, a medium conditioned by GPCs may be promising not only for TBI therapy but for other neurological diseases. However, the promotion of the PC medication in the clinic needs further study, including the dose-dependent effects and product standardization.

## Figures and Tables

**Figure 1 ijms-24-12341-f001:**
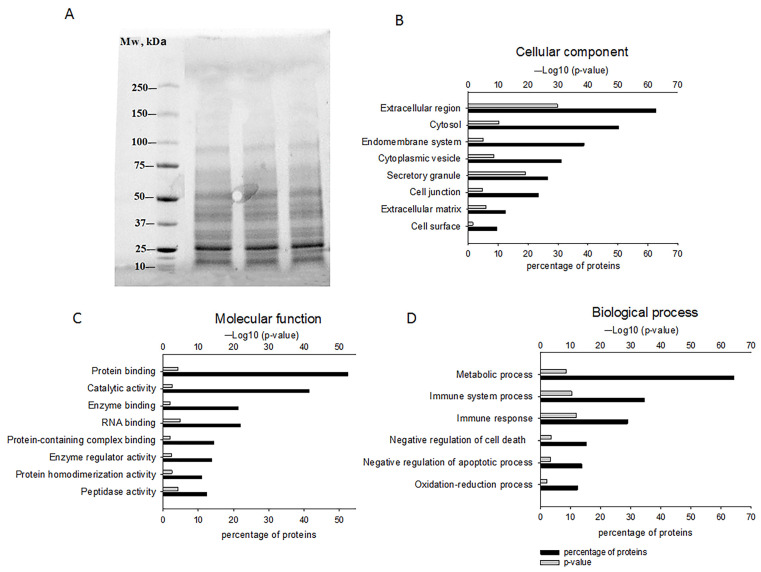
The characterization of GPC-secreted proteins. (**A**) Gel electrophoresis results reflecting the presence of proteins below 100 kDa. Proteomic analysis of identified proteins and their classification into cellular components (**B**), molecular functions (**C**), and biological processes (**D**) using https://string-db.org/ (accessed on 20 July 2023).

**Figure 2 ijms-24-12341-f002:**
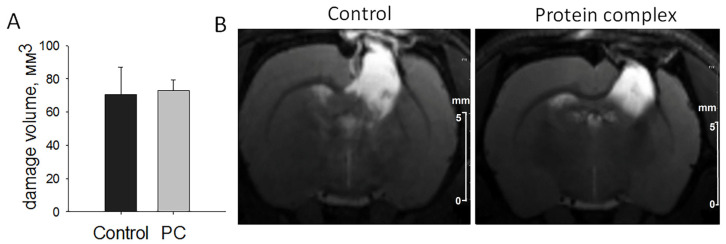
The assessment of damage volume by MRI on day 14. (**A**) Comparison of the quantitation data presented as means ± SD. (**B**) Representative T2-weighted images, scale bars, 5 mm.

**Figure 3 ijms-24-12341-f003:**
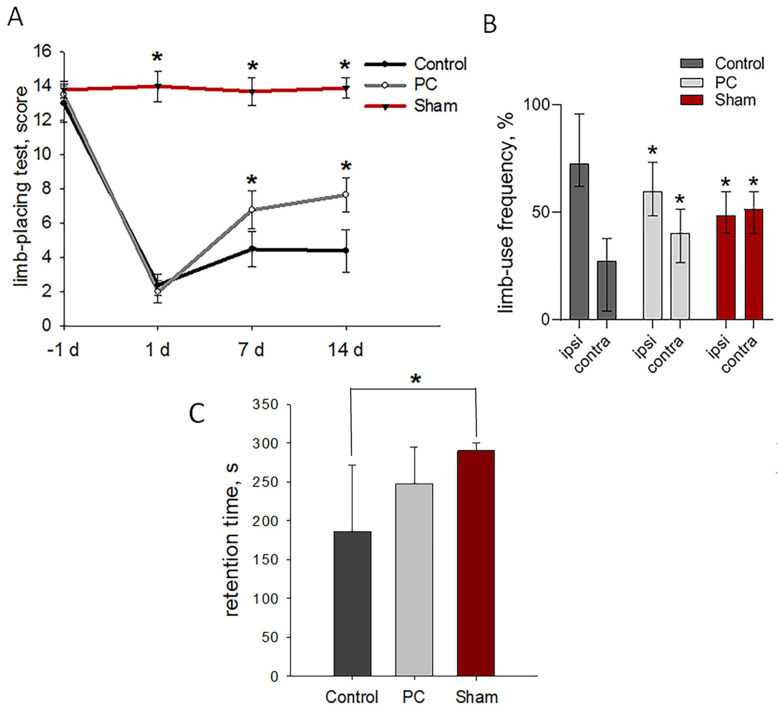
Neurological outcomes of animals during post-TBI recovery. (**A**) Limb-placing test: the data are presented as means ± SD, * *p* < 0.05 by two-way ANOVA, with Holm–Sidak test for multiple comparisons. (**B**) Cylinder test of forelimb use asymmetry: the data are presented as median with range, * *p* < 0.05 by ANOVA on ranks with Dunn’s post-hoc test; ipsi—ipsilateral limb and contra—contralateral limb. (**C**) Rotarod test: the data are presented as means ± SD, * *p* < 0.05 by one-way ANOVA, with Holm–Sidak test for multiple comparisons.

**Figure 4 ijms-24-12341-f004:**
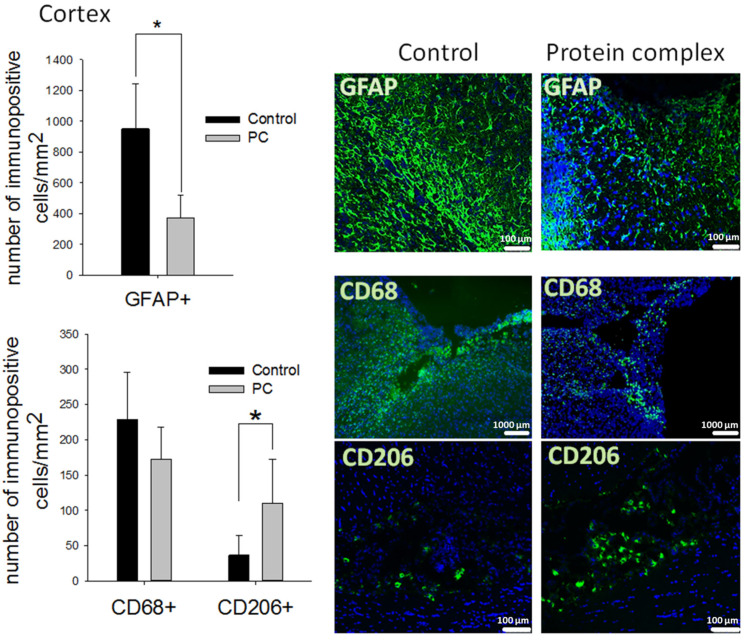
Immunohistochemical study of the brain on day 14 post-TBI. Quantitative assessment of astrocytes (GFAP^+^ cells), phagocytic macrophages (CD68^+^ cells), and M2 macrophages (CD206^+^ cells) at the site of cortex injury, and representative images. The data are presented as means ± SD; * *p* < 0.05 (*t*-test). Cell nuclei counterstained with DAPI (blue); scale bars, 100 and 1000 µm.

**Figure 5 ijms-24-12341-f005:**
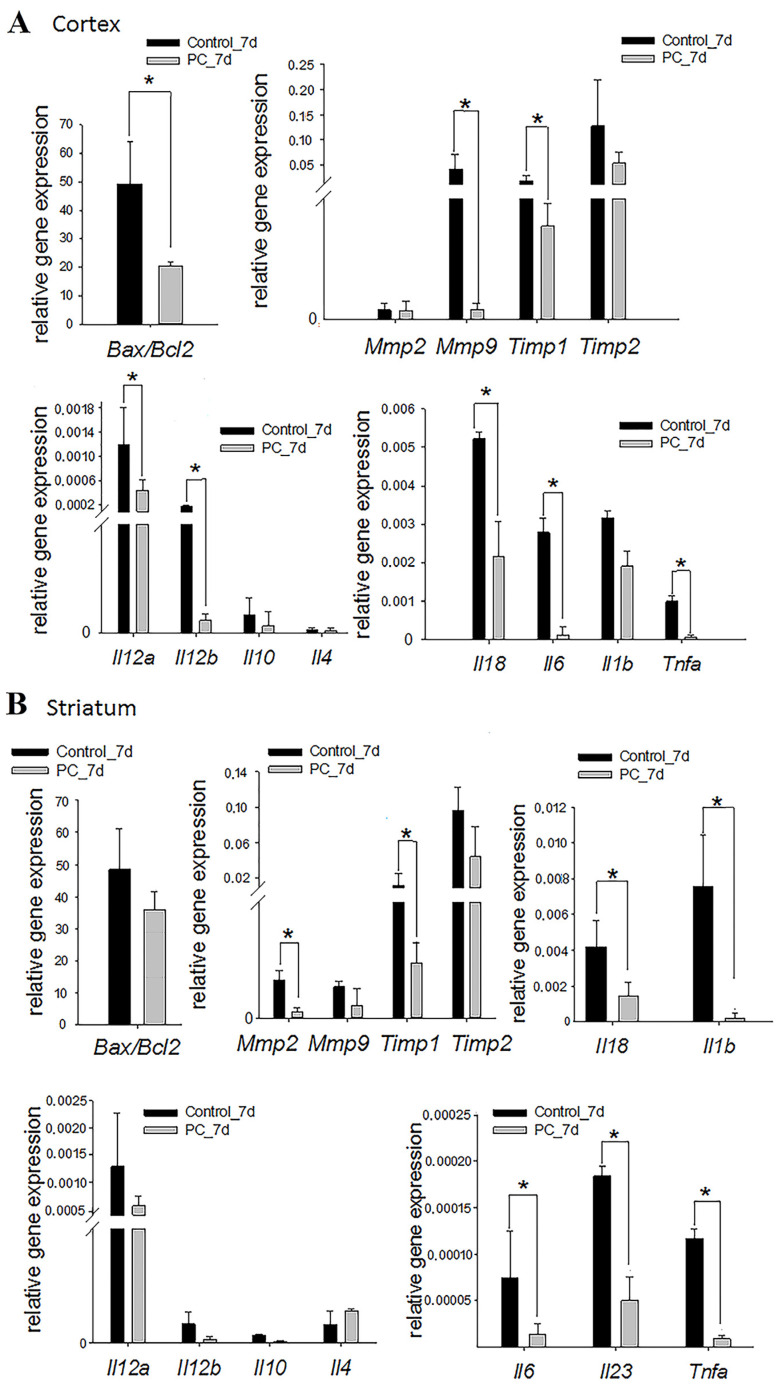
Expression levels of the apoptosis, extracellular matrix degradation, and inflammatory markers at the cortex injury site (**A**) and in the striatum (**B**) on day 7 post-TBI. The RT-PCR data are presented as means ± SD * *p* < 0.05 (*t*-test or Mann–Whitney test).

**Figure 6 ijms-24-12341-f006:**
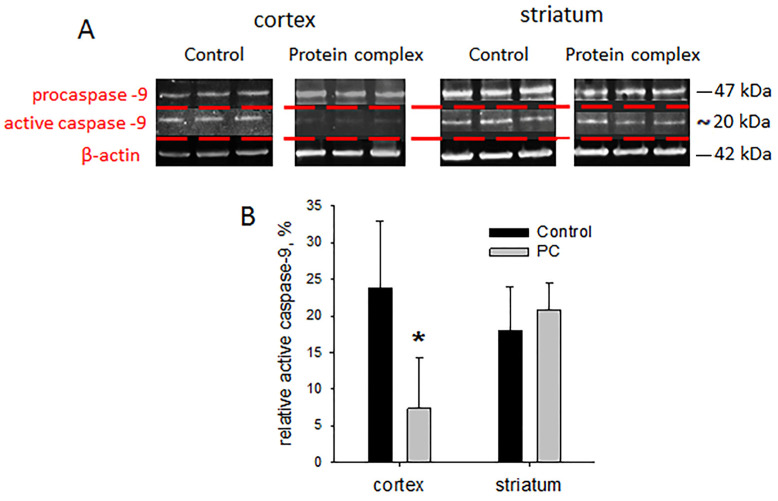
Immunoblotting for caspase-9: (**A**) representative blots, (**B**) semi-quantitation data presented as means ± SD; * *p* < 0.05 (*t*-test).

**Figure 7 ijms-24-12341-f007:**
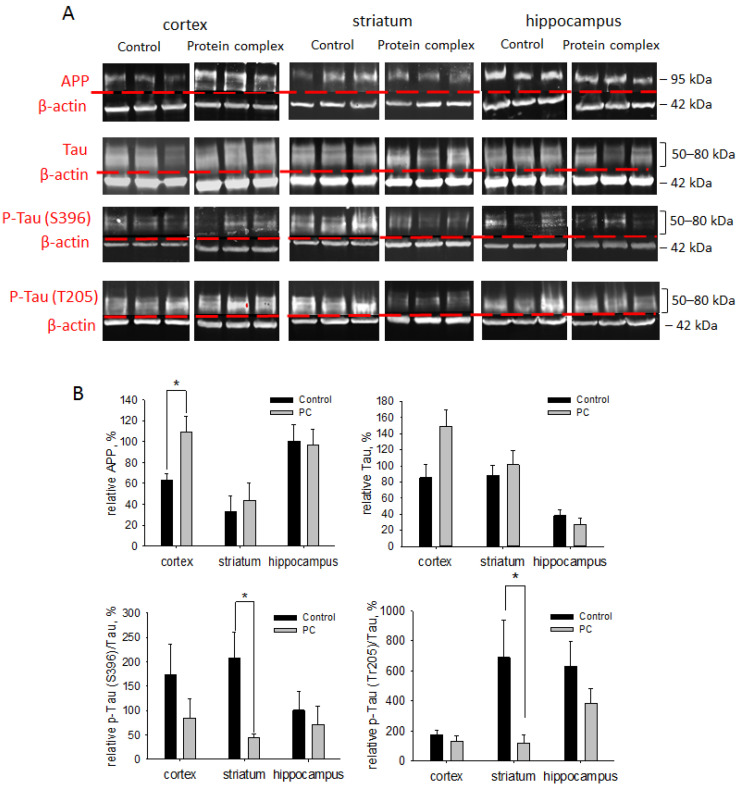
Immunoblotting for neurodegeneration-associated proteins APP, Tau, and p-Tau proteins: (**A**) representative blots; (**B**) semi-quantitation data presented as means ± SD; * *p* < 0.05 (*t*-test).

**Figure 8 ijms-24-12341-f008:**
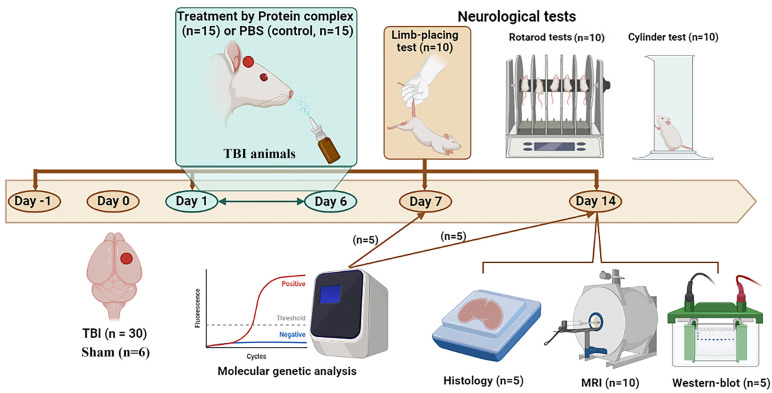
Design of the experiments depicted using https://biorender.com/ (accessed on 31 July 2023). TBI, traumatic brain injury; MRI, magnetic resonance imaging.

## Data Availability

All data collected or analyzed during this study are included in this article and its Appendix A.

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
