# Peer review of "Therapeutic Efficiency of Proteins Secreted by Glial Progenitor Cells in a Rat Model of Traumatic Brain Injury"

_ijms, 2023, doi:10.3390/ijms241512341_

Round 1
Reviewer 1 Report
The authors came up with an interesting finding in intranasal PC infusions for treating TBI.
I have a few minor comments for improving the quality of the manuscript.
1. Keep consistent abbreviations all over the manuscripts, I have observed MW and Mw for molecular weight.
2. PC abbreviation was used before it was mentioned in the manuscript.
3. Quality of figures need to be improved such as legends, images, and scales
4. Figure 4, the authors showed CD68 and CD206 IHC images, my question is why do both control and PC images show predominant CD68 signals all over the tissue? Is this a background signal? .Also, CD206 signals look like in blood vessel staining ( we should not consider false signals).
5. In figure 6, I can not see the band for tau antibodies besides noise.
6. Discussion needs to be improved
Author Response
Thanks for the comments that helped improve the article. We took into account all the comments and the changes are highlighted in the yellow color.
- Keep consistent abbreviations all over the manuscripts, I have observed MW and Mw for molecular weight.
- We corrected all abbreviations in the manuscript.
- PC abbreviation was used before it was mentioned in the manuscript.
- We added the PC abbreviation at its first mention.
- Quality of figures need to be improved such as legends, images, and scales
- We have improved all figures. In addition, we have added new figure. In order to show the anti-apoptotic effect of PC, besides the changing of the expression level of the Bax and Bcl2 genes, we studied the level of caspase-9 by Western Blotting (new Figure 6).
- Figure 4, the authors showed CD68 and CD206 IHC images, my question is why do both control and PC images show predominant CD68 signals all over the tissue? Is this a background signal? .Also, CD206 signals look like in blood vessel staining ( we should not consider false signals).
- Figure 4 shows an area of cortical damage, and CD68+ monocytic macrophages are present throughout this area (this is not a background signal), since the area of damage is quite extensive. Indeed, immunohistochemistry on CD206 macrophages looks ambiguous, so we replaced it with another photo. We using these antibodies (CD206, ab64693) for a long time and have demonstrated their specificity (details can be found here https://doi.org/10.3390/biomedicines8120627 ). Moreover, some studies have shown the localization of CD206 macrophages surrounding blood vessels at the periphery and core of brain damage area (https://doi.org/10.1016/j.nbd.2019.104722). Probably due to the perivascular location of CD206 macrophages, the signals looked like staining of blood vessels.
- In figure 6, I can not see the band for tau antibodies besides noise.
- We remade the blots for Tau protein, which made it possible to demonstrate the level of this protein in a better quality. This figure has been modified in accordance with the comment (now it is figure 7).
- Discussion needs to be improved
- We have substantially revised the discussion.
Reviewer 2 Report
Diana I.Salikhova et al. studied the potential therapy of secreted proteins from cultured GPC in the TBI model. The idea is novel and innovative. However, I have several major comments as below.
The uninjured control is missing and it's hard to tell how much PC treatment improves or worsens the outcomes.
The intranasal administration method can’t deliver the protein to the injury site directly. So how to ensure the protein reaches the injury site and what forms of components finally function at the injured site?
GPC complex has numerous proteins, but the authors didn’t show which ones play the role of protection.
The logic behind choosing GPC-secreted protein is not clarified. GPC’s role in TBI should be mentioned to introduce the design in the present study.
The damage is extensive according to the MRI image, more neurological disruption could be observed in addition to the sensorimotor deficit. This could help determine the wider effect of GPC proteins.
The time-points in Figures 4 and 5 are inconsistent. The unit in Figure 5 is so different, needs to keep consistent.
Figure 6 images of better quality need to be presented. The order of A and B should be reversed. The black bands were more commonly used.
Author Response
Thanks for the comments that helped improve the article. We took into account all the comments and the changes are highlighted in the yellow color.
- The uninjured control is missing and it's hard to tell how much PC treatment improves or worsens the outcomes.
- We have added a sham-operated group (see figure 3).
2 . The intranasal administration method can’t deliver the protein to the injury site directly. So how to ensure the protein reaches the injury site and what forms of components finally function at the injured site?
- Many studies have shown that the method of intranasal administration makes it possible to deliver the protein to certain parts of the brain. Besides, intranasal administration appreciated as a method for direct delivery of therapeutics protein to the CNS, effectively bypassing the BBB (https://doi.org/10.3109/10717549709051878, doi: 10.2174/1567201053586047 , https://doi.org/10.1517/17425247.2011.588204). Intranasal delivery some factors (for example, BDNF, CNTF, EPO, and NT-4) at 25 min can achieve brain regions: superior colliculus, Inferior colliculus, thalamus, hypothalamus, hippocampus, frontal cortex, parietal cortex, occipital cortex, olfactory bulbs, trigeminal nerve (doi: 10.3109/10611860903318134). All these data suggested that many proteins can achieve the cortical and subcortical regions via intranasal administration. Thus, protein complex can reaches the injury site. According bioinformatics analysis about 30% identified proteins regulate immune system processes and immune response, and 13–15% relate to negative regulation of cell death/apoptosis. These proteins function (reducing the levels of apoptosis and inflammation at the injured site) were demonstrated by intranasal administration PC. But it is difficult to define which certain component finally functions at the injured site because of the multicomponent of secretome, and proteins with proven neuroprotective and anti-inflammatory actions are not one in the PC, but several.
- GPC complex has numerous proteins, but the authors didn’t show which ones play the role of protection.
- In results section chapter 2.1. «The composition of GPCs secreted proteins» was described which proteins, in our opinion, play a key role in TBI. Also in the discussion section we have added information about proteins that may be associated with the observed effects of РС.
4. The logic behind choosing GPC-secreted protein is not clarified. GPC’s role in TBI should be mentioned to introduce the design in the present study.
- The GPC’s role in TBI was added in discussion section. Line 276-283.
5. The damage is extensive according to the MRI image, more neurological disruption could be observed in addition to the sensorimotor deficit. This could help determine the wider effect of GPC proteins.
- We appreciate your ideas and would like to respond to your comments regarding the modeling of damage to the sensorimotor cortex. The primarily damage affects the sensorimotor area of the cortex, secondary damage can occur other structures, such as hippocampus and striatum, due to systemic inflammation. By demonstrating the restoration of sensorimotor functions that were directly damaged in the area of the sensorimotor cortex, we sought to provide evidence of a broader potential impact of our therapeutic interventions - studying the effects of PC on the hippocampus and striatum. In addition, our results indicate a high neuroprotective effect of PC, as evidenced by the observed restoration of sensorimotor functions. This indicates that the therapeutic approaches we are studying can provide protection not only to the directly damaged sensorimotor cortex, but also to other areas of the brain affected by secondary damage. We hope that this will clarify our point of view and the rationale for focusing on sensorimotor testing and the subsequent impact on the restoration of brain functions.
6. The time-points in Figures 4 and 5 are inconsistent. The unit in Figure 5 is so different, needs to keep consistent.
-Units in Figure 5 were corrected. Molecular genetic analysis was assessed on days 7 and 14 post-TBI, and immunohistochemistry analysis only at the end point of the experiment (day 14) to minimize the number of animals. At the same time, changes in gene expression are mainly observed in the acute phase of TBI (day 7), but not in the chronic phase (day 14, see Figures S3B and S4 supplementary material), which can be explained by the administration of PC in the acute phase of TBI. Therefore, the time-points in Figures 4 and 5 are differing.
7. Figure 6 images of better quality need to be presented. The order of A and B should be reversed. The black bands were more commonly used.
-This figure has been modified in accordance with the comments (now it is figure 7). Moreover, we remade the blots to Tau protein, which made it possible to demonstrate the level of this protein in a better quality. In order to show the anti-apoptotic effect of PC, besides the changing of the expression level of the Bax and Bcl2 genes, we studied the level of caspase-9 by Western Blotting (now it is Figure 6).
Round 2
Reviewer 2 Report
Thanks for the response.
The responses have addressed all the comments.